# Climatic change and extinction risk of two globally threatened Ethiopian endemic bird species

Andrew J. Bladon[1]*, Paul F. Donald[1,2,3], Nigel J. Collar[2], Jarso Denge[4], Galgalo Dadacha[4], Mengistu Wondafrash[5], Rhys E. Green[1,3]

1 Conservation Science Group, Department of Zoology, University of Cambridge, Cambridge, United Kingdom, 2 BirdLife International, Cambridge, United Kingdom, 3 RSPB Centre for Conservation Science, RSPB, The Lodge, Sandy, Bedfordshire, United Kingdom, 4 Borana National Park Authority, Yabello, Oromiya, Ethiopia, 5 Ethiopian Wildlife and Natural History Society, Bole Sub City, Addis Ababa, Ethiopia

* andrew.j.bladon@gmail.com

**Data Availability Statement:** The data are available here: https://doi.org/10.17863/CAM.65907.

**Funding:** AJB was supported by a PhD studentship grant from the Natural Environment Research

## Abstract

Climate change is having profound effects on the distributions of species globally. Trait-based assessments predict that specialist and range-restricted species are among those most likely to be at risk of extinction from such changes. Understanding individual species' responses to climate change is therefore critical for informing conservation planning. We use an established Species Distribution Modelling (SDM) protocol to describe the curious range-restriction of the globally threatened White-tailed Swallow (*Hirundo megaensis*) to a small area in southern Ethiopia. We find that, across a range of modelling approaches, the distribution of this species is well described by two climatic variables, maximum temperature and dry season precipitation. These same two variables have been previously found to limit the distribution of the unrelated but closely sympatric Ethiopian Bush-crow (*Zavattariornis stresemanni*). We project the future climatic suitability for both species under a range of climate scenarios and modelling approaches. Both species are at severe risk of extinction within the next half century, as the climate in 68–84% (for the swallow) and 90–100% (for the bush-crow) of their current ranges is predicted to become unsuitable. Intensive conservation measures, such as assisted migration and captive-breeding, may be the only options available to safeguard these two species. Their projected disappearance in the wild offers an opportunity to test the reliability of SDMs for predicting the fate of wild species. Monitoring future changes in the distribution and abundance of the bush-crow is particularly tractable because its nests are conspicuous and visible over large distances.

## Introduction

The effects of climate change on the distribution and abundance of animal and plant species are well documented. They include range shifts and changes in local density, phenology, morphology, behaviour and gene frequencies [1–5]. Because of long-term monitoring of bird distributions and population densities in north temperate regions, the best studied of these effects are range shifts and population changes of European and North American birds [4–6]. Range

Council (award 1210186), the Royal Society for the Protection of Birds, British Birdfair/RSPB Fund for Endangered Species, African Bird Club Expedition Awards, British Ornithologists' Union Small Grants Scheme, Cambridge Philosophical Society, Department of Zoology Tim Whitmore Fund, Cambridge, Worts Travelling Scholars Fund, Cambridge, and Magdalene College, Cambridge. The funders had no role in study design, data collection and analysis, decision to publish, or preparation of the manuscript.

**Competing interests:** The authors have declared that no competing interests exist.

shifts in response to climate warming are characterised by two processes, "cold-edge" expansion and "warm-edge" contraction, which begin with increases and decreases in local density at the two edges respectively [7]. At the warm-edge of species' ranges, limitations imposed on foraging behaviour, breeding success or survival by rising temperatures and/or associated drought can lead to local population declines [8–13]. Meanwhile, at the cold-edge, the removal of the lower thermal limit on these processes enables local populations to thrive [6, 14, 15].

However, focus on poleward shifts alone may underestimate the impacts and complexity of climate change, especially in the tropics, where responses of birds are less well known. One meta-analysis suggests that climatic effects on population processes of birds are more likely to involve temperature in temperate regions, and precipitation or aridity in the tropics [3]. In Africa, species distribution modelling has predicted multidirectional range contraction, with distributions of southern bird species projected to become more restricted in the Cape Region, and inhabitants of other regions, including the Horn of Africa, projected to decrease their range size, particularly as arid areas expand [16]. In South Africa, two Fynbos endemics, Cape Rockjumper (*Chaetops frenatus*) and Protea Canary (*Serinus leucopterus*), whose distributions are limited by temperature, have suffered reductions of over 30% in both range extent and reporting rates since the late 1980s, consistent with a loss of potential range predicted by recent climate change and climate envelope models [17].

Without remedial conservation action, persistent "warm-edge" declines which outpace any "cold-edge" expansion will eventually lead to population extinctions. A meta-analysis of model-based predictions of extinction rates from climate change varied greatly among studies [18–21], ranging from 0% to 54% of species, with a mean of 7.9% [22]. Much variation in these predictions is associated with baseline data of different types and quality and with variation in the climate change scenarios and global circulation models (GCMs) used [23–25], but the overall pattern across studies is for predicted population declines and range contractions to predominate over increases and expansions at a global level [5]. In the face of ongoing climatic change, and the lagged effects which may accrue, many species could already be committed to extinction by 2050 [18]. For species that exhibit direct physiological intolerance of high temperatures [12, 13], climate change also threatens to increase the frequency and severity of episodes of high mortality caused by heatwave events [26], even in areas where average climatic conditions remain suitable [27].

The likely severity of climate change impacts on species' populations has been assessed by using postulated effects of the ecological and life-history traits of a species on its sensitivity, exposure and capacity to adapt to climate change [20]. Using this approach, Foden et al. [20] assessed the family to which the one of the focal species of our study, the White-tailed Swallow (*Hirundo megaensis*) belongs (Hirundinidae; swallows and martins) as being the least vulnerable to climate change of all bird families [20]. However, individual species with restricted ranges and narrow environmental tolerance are likely to be particularly susceptible to climate change [12, 16, 28]. Assessing the projected impacts of local climate change on individual species is, therefore, important for assessing their long-term conservation prospects.

Species Distribution Models (SDMs) have demonstrated that the peculiarly restricted distribution of the Ethiopian Bush-crow (*Zavattariornis stresemanni*) is well described by a climate envelope model, encompassing a zone of cooler, drier conditions than surrounding areas [12, 29]. The apparent range limitation by maximum temperature may be linked to the effects of ambient temperature on thermoregulatory and foraging behaviour [12]. The small, non-migratory White-tailed Swallow has a global distribution very similar to that of the bush-crow [30, 31]. Several authors have noted its peculiarly restricted distribution [32–34], which Collar and Stuart [35] suggested might be linked to the 1,500 m altitudinal contour [35]. Since 2006, there have been records from outside the previously known breeding range, 100 km to the east

on the Liben Plain [36], but there are still no nest records from this region, and it is unclear whether individuals move between the two areas [37]. It seems possible, therefore, that the range of the White-tailed Swallow might be restricted by similar attributes of the local climate, albeit probably by a different mechanism given its very different ecology, phylogeny and phenotype.

Models that successfully predict the current distribution of a species using a small number of bioclimatic variables can be used to predict the potential future range under different projected climate scenarios [3, 16]. These predictions can be used to target areas for habitat protection and restoration which offer the best hope as thermal refugia for temperature-sensitive globally-threatened species [38]. Both the Ethiopian Bush-crow (Endangered) and the White-tailed Swallow (Vulnerable) are categorised as globally threatened in the IUCN Red List [31, 39]. Understanding the potential impact of climate change on the range boundaries of the bush-crow and swallow is necessary to develop conservation management plans for them in the newly formed Yabello National Park. In this paper, we fit SDMs to the small global range of the White-tailed Swallow, using the same techniques as for the bush-crow [12]. We then combine the SDMs for both species with projected future climate scenarios for the region, to predict how their potential climatic range is likely to be affected by changes in temperature and precipitation in the future.

## Materials and methods

### Modelling the current distribution of the White-tailed Swallow

We collated all available geo-referenced records of White-tailed Swallows and their nests, collected by various observers between 2005 and 2011 [30, 36, 37], including sightings made during fieldwork on the bush-crow [29]. Nests are usually built in the traditional huts occupied by local people and take the form of mud cups typical of the genus *Hirundo*, but in the absence of adult birds can be distinguished from sympatric hut-nesting swallows by their simple grass lining and unmarked white eggs [37]. Between 2012 and 2015, we conducted 255 walked 1-km transects at locations across, and outside, the species' core range (see [12] for further details). White-tailed Swallows were recorded on 19 (7.5%) of these transects. Additionally, in 2014, nest records were documented in the north-west of the species' range as part of an intensive breeding study [40]. We also collected GPS locations for all ad-hoc White-tailed Swallow observations throughout this period, including from the Liben Plain. This work was carried out under permit from the Ethiopian Wildlife Conservation Authority.

We previously fitted SDMs for the Ethiopian Bush-crow [12]. Models for the White-tailed Swallow were built using the same five climatic variables from WorldClim [41]—maximum temperature of the warmest month, temperature seasonality, annual temperature range, precipitation of the wettest quarter, and precipitation of the driest quarter—using the same SDM procedures from the R package 'biomod2' [42]. We used all 574 records of the White-tailed Swallow as presence data. For absence data, we took the mid-point of the 236 1-km transect legs on which swallows had not been recorded. Unlike the models for the bush-crow, we did not use the lack of observations from stretches of road transects as true absence points because, unlike the bush-crow and its highly visible nests, White-tailed Swallows are much more difficult to detect reliably from a moving vehicle [12]. This is because a) there are a number of other swallow species found in the area [43], making positive identification from a moving vehicle unreliable, b) White-tailed Swallows are small and often occur singly or in pairs [37] and c), if they are like other swallow species, they are likely to forage over a large area and may congregate in areas with plentiful food. These things all make it unreliable to assume that the failure to detect them at a particular place from a moving vehicle denotes a true absence of the

species from the surrounding area. To increase the range of environmental variables on which the models were built, we took a random sample of 4,764 pseudo-absences from a region stretching from 1.86˚–6.87˚N and 33.17˚–43.67˚E. We chose this extent for consistency with previous studies [12, 29], and because it represents a pragmatic compromise between choosing an area large enough to ensure a range of environmental variables extending beyond the species' known distribution–which is important for making predictions based on possible future scenarios–but small enough to make the models biologically relevant to a species with such a restricted range [44]. We restricted these points to be at least 10 km from any presence location. When combined with the 236 transect-based absence records, this gave a total of 5,000 points treated as absences in the analysis for consistency with the bush-crow models [12].

For model validation purposes, the White-tailed Swallow's range was divided radially into five geographic segments (Fig 1). We fitted SDMs using seven model algorithms—Generalised Linear Models (GLMs), Generalised Additive Models (GAMs), Flexible Discriminant Analysis (FDA), Multiple Adaptive Regression Splines (MARS), Boosted Regression Trees (BRT), Random Forests (RF) and Maximum Entropy (MaxEnt)—and assessed the ability of each model to predict the swallow's current distribution using a $k$-fold leave-one-out cross-validation (LOOCV) method [12, 45]. We fitted each model five times, leaving out the data from one of the five radial segments in each case. The fitted model was then used, with the bioclimate variable values, to predict probability of occurrence for each presence/absence location within the segment whose data had been omitted. Having used this $k$-fold LOOCV approach to make

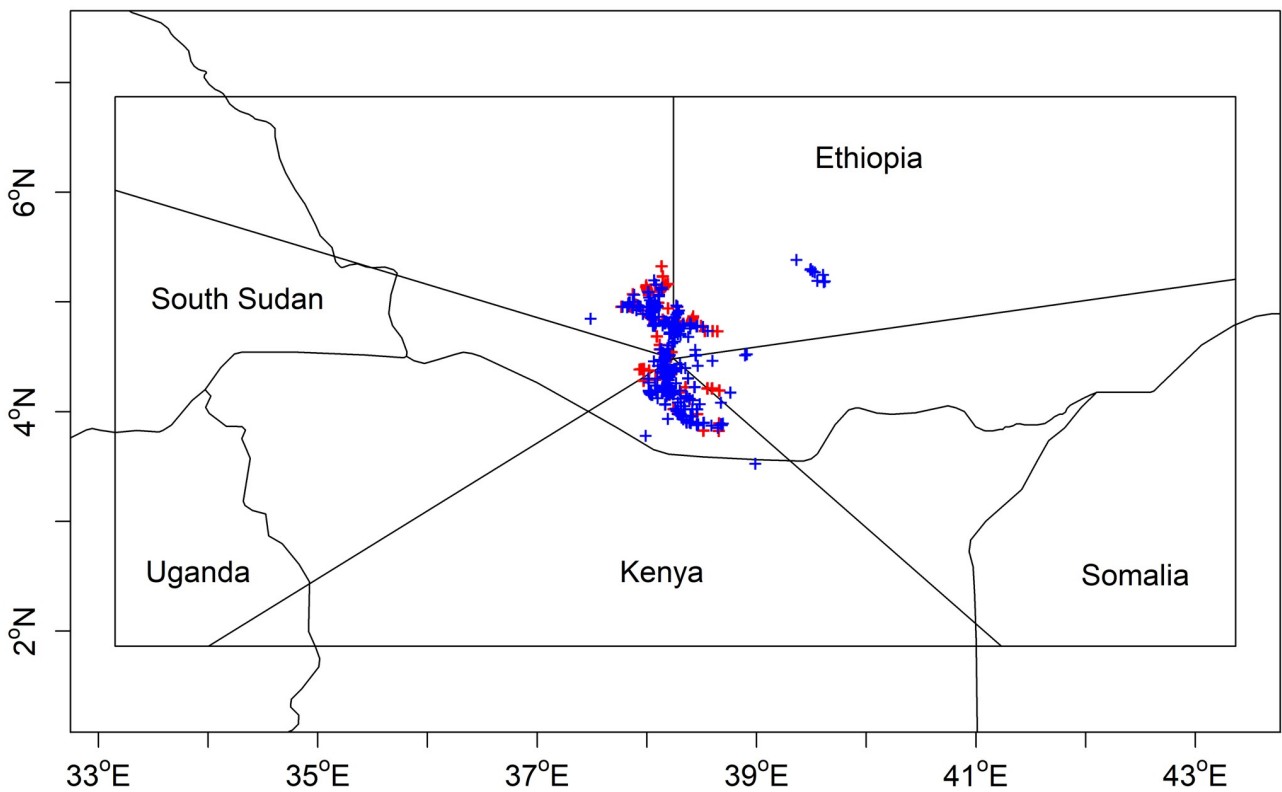

**Fig 1. Global range map for the White-tailed Swallow.** The co-ordinates of all presence (blue) and transect-based absence (red) locations used for fitting species distribution models are shown. The rectangular box shows the area across which pseudo-absence locations were drawn for modelling, and current and future simulations were projected. The lines radiating from the centre show the five sectors of the map used for leave-one-out cross-validation of the models. International borders are plotted using the '*wrld_simpl*' dataset available in the '*maptools*' package in R [47, 48].

predictions for sites in all five segments, we then used the predictions and the observed presence/absence data for all segments to calculate the Area Under the Receiver Operating Characteristic Curve statistic (AUC-ROC) [3, 46].

To assess variable importance, we selected results from the model-fitting algorithms with k-fold LOOCV AUC scores greater than 0.6 (MaxEnt, GLM and GAM). Although scores exceeding 0.7 are preferred [49], none of our models obtained this threshold. Since these three models had similar k-fold LOOCV AUC scores (see Results), we assessed variable importance and future climatic suitability based on all three models to avoid biasing our results towards a single model algorithm [24, 25]. We re-fitted each of these models, using the LOOCV procedure, with each variable left out in turn and calculated the k-fold LOOCV AUC statistic for each of these models. Delta AUC scores were then calculated by subtracting the k-fold LOOCV AUC for the models with the variable missing from the k-fold LOOCV AUC for the model with all bioclimate variables included. In order to compare variable importance between the swallow and the bush-crow, we standardised these scores by dividing the delta AUC score for each variable by the sum of the delta AUC scores from all five variables in the model [12].

## Projecting future climatic suitability for the Ethiopian Bush-crow and White-tailed Swallow

In addition to the recent climatic data we used for model building, the WorldClim database [41] contains future projections of the same bioclimatic variables from a range of GCMs and Representative Concentration Pathways (RCPs) [50, 51]. We obtained projections of annual values of bioclimate variables from WorldClim for six GCMs (S1 Table in S1 File) and all four RCPs for which results are available [41]. We calculated projected average values for each bioclimate variable in two years: 2050 (by averaging projections for the period 2041–2060) and 2070 (average for 2061–2080) (S2 Table in S1 File). We combined these projected bioclimate values with our SDMs, built under current conditions, to project the potential range of both species under different scenarios from the expected probability of occurrence for each 30 arc-second grid cell.

The four RCPs considered (RCP2.6, RCP4.5, RCP6.0, RCP8.5) cover a range of possible radiative forcing values for 2100, from 2.6 to 8.5 W/m² [51], taking account of multi-gas emission scenarios. The lowest emissions scenario, RCP2.6, assumes that atmospheric greenhouse gas concentrations peak before 2050 and decline thereafter, while the other three scenarios assume progressively higher and later stabilisation of greenhouse gas concentrations. The best-estimate global mean surface temperature increases are respectively 0.3–1.7˚C, 1.1–2.6˚C, 1.4–3.1˚C and 2.6–4.8˚C by 2100, relative to the mean of 1986–2005 [50]. Together the four RCPs represent the range of scenarios considered plausible by the Intergovernmental Panel on Climate Change [52].

The choice of GCM and modelling technique can significantly impact climate change predictions, and the effect increases with distance of future projections, tending to outweigh differences arising from the initial dataset used or climate change scenario (i.e. RCP) selected [24, 25]. However, projections of future range changes are more consistent for species with restricted environmental niches like the bush-crow and swallow [24].

We used k-fold LOOCV AUC comparison of SDMs built using current climate data for the Ethiopian Bush-crow [12] and White-tailed Swallow to select the model algorithms which produced the highest AUC scores when projected over each species' current distribution (BRT and RF for the bush-crow, and MaxEnt, GLM and GAM for the swallow). We then refitted these models using all available data (i.e. not using the LOOCV procedure), and projected the

results under the six GCMs, four RCPs and for two projection years (2050 and 2070) [41] to assess the potential impacts of climate change on the two species.

To determine the projected future range sizes of the two species, we first calculated maximum kappa for the current range simulations according to each model algorithm. We used the probability of occurrence threshold which yielded maximum kappa under current conditions to convert the probability of occurrence for each future scenario into binary presence–absence scores in each cell, and summed the area of the cells where the species' presence was predicted. We took the mean area across all six GCMs and the projected model algorithms for each species, to produce the mean potential climatically suitable area under each RCP/time-period scenario. By subtracting the remaining climatically suitable area under each scenario from the current simulated range size (based on maximum kappa) for the best-fitting models, we calculated the mean percent loss of climatically suitable range under each scenario for each species.

## Results

### Modelling the current distribution of the White-tailed Swallow

SDMs fitted for the White-tailed Swallow found that three model algorithms performed best: Maximum Entropy (MaxEnt, $k$-fold LOOCV AUC score = 0.627), Generalised Linear Models (GLM, $k$-fold LOOCV AUC score = 0.619) and Generalised Additive Models (GAM, $k$-fold LOOCV AUC score = 0.601). These scores were much lower than the best-performing model for the Ethiopian Bush-crow (BRT, $k$-fold LOOCV AUC score = 0.824; [12]). Precipitation in the driest quarter produced the highest delta AUC score under each algorithm (MaxEnt = 0.146, GLM = 0.047, GAM = 0.089), followed by maximum temperature of the warmest month (MaxEnt = 0.121, GLM = 0.040) or precipitation in the wettest season (GAM = 0.038; Fig 2, S3 Table in S1 File). Response plots indicated well-defined dry-season rainfall (50–70 mm) and maximum temperature (30–35˚C) thresholds, above which White-tailed Swallows did not occur (Fig 3, S1 Fig in S1 File). The GLM and GAM models predicted White-tailed Swallow occurrence across a slightly wider range of dry-season rainfall values, and at slightly higher temperatures, than did the MaxEnt model (Fig 3).

### Projecting future climatic suitability for the Ethiopian Bush-crow and White-tailed Swallow

Projections of future bioclimate values within the current range of both the swallow and the bush-crow indicated an increase in maximum temperature beyond the threshold at which the two species currently occur, while there was less projected change in precipitation (Fig 4, S2 Table in S1 File). There was some variation between GCMs in the location and size of the predicted potential range of each species for a given RCP and time period. However, a severe future decline in projected suitable area was observed across RCPs under each GCM, primarily caused by rising temperature (Figs 5 and 6). Under all scenarios, both species' potential ranges are projected to contract markedly, in some cases leading to a total loss of suitable area by 2070.

Summarised across models, and depending on which RCP is realised, the species are projected to lose 85–96% (bush-crow) and 56–79% (swallow) of potential range by 2050, and 90–100% (bush-crow) and 68–84% (swallow) by 2070, relative to the current mean climatically suitable area according to the best climate-only models for each species ([12]; Table 1). Such decreases will leave remaining areas which are likely to be too small to support viable populations.

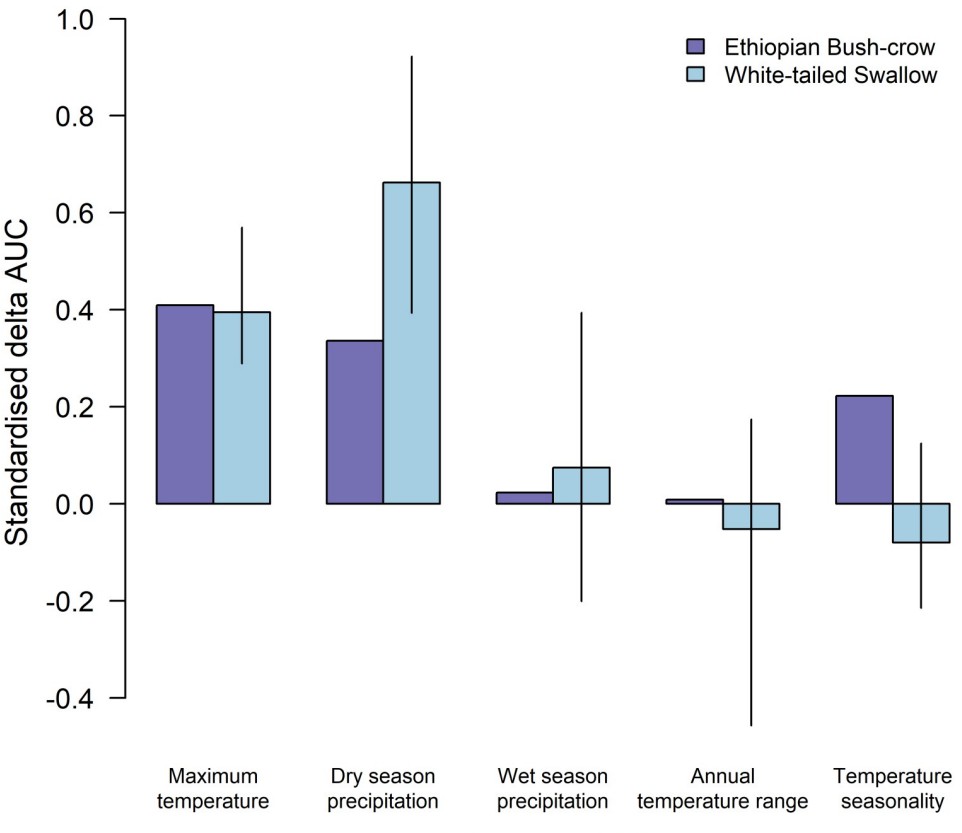

**Fig 2. Comparison of the relative importance of five bioclimate variables in models describing the geographical range of the Ethiopian Bush-crow (taken from [12]) and White-tailed Swallow.** For the bush-crow, bars represent standardised delta AUC scores from the model fitted using the algorithm which gave the best fit (boosted regression trees). For the swallow, bars represent the mean standardised delta AUC scores from the three best-performing models (Maximum Entropy, Generalised Linear Models, Generalised Additive Models), which were indistinguishable based on their *k*-fold LOOCV AUC scores. Lines represent the range of standardised delta AUC scores from the three algorithms for the swallow.

## Discussion

Like the Ethiopian Bush-crow [12, 29], the global distribution of the White-tailed Swallow is closely correlated with aspects of the local climate, being drier and cooler within the range edge than outside it. The mechanism by which two such unrelated species, with very different behaviour and ecology and with no evidence of an interdependent relationship, have come to have such similar ranges, apparently defined by the same climatic variables, is unknown. The habitats used by the two species are similar, and consist of a mixture of rather degraded savanna scrub and open grassland with well-spaced trees. Habitats in areas adjacent to the species' ranges appear to be remarkably similar to those within it, and the fit of models of bush-crow distribution was not markedly improved when habitat variables were included [29, 40]. We are not aware of any species in other taxonomic groups that share these two species' patterns of distribution. For the bush-crow, range restriction is explained, at least in part, by the inability of birds to forage efficiently at temperatures above its climatic limit, because of the thermoregulatory need to pant and seek shade [12]. For the swallow, the mechanism constraining it is unknown, but may be mediated by a decline in breeding success at higher ambient temperatures [40].

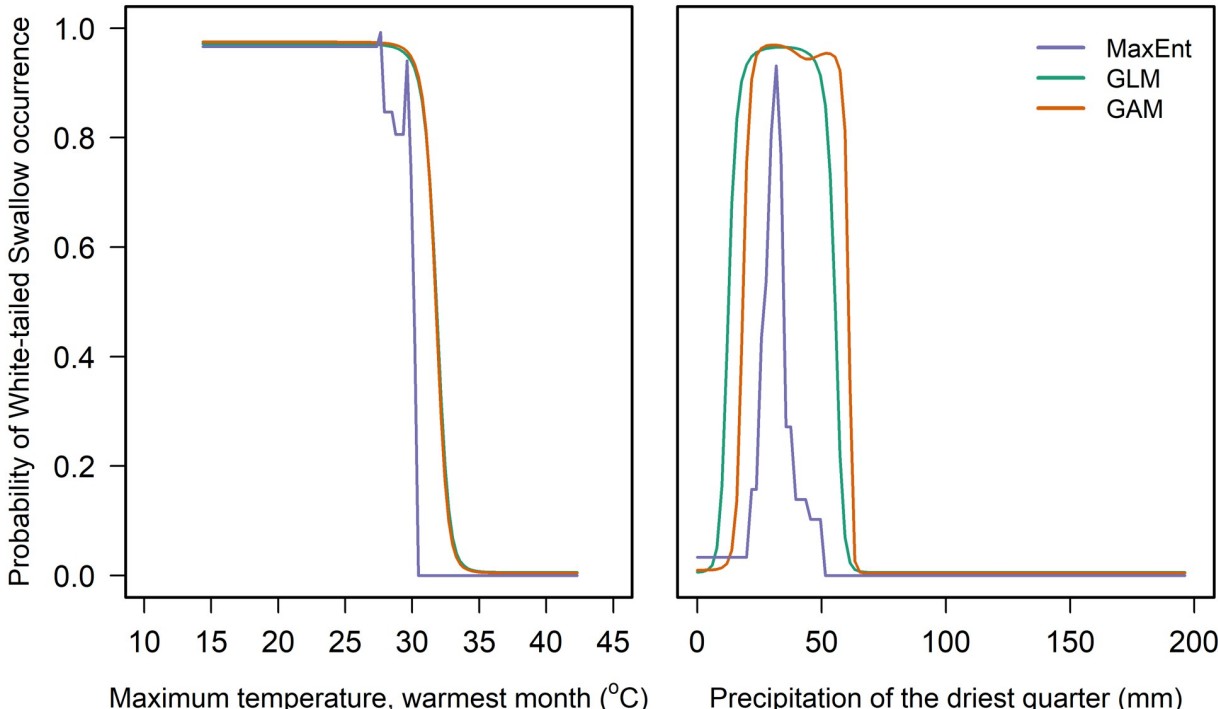

**Fig 3. The relationship of modelled partial probability of occurrence of White-tailed Swallow to (left) maximum temperature of the warmest month and (right) precipitation of the driest quarter.** All other bioclimate variables are held constant. Curves show the predicted responses determined using the three model algorithms (Maximum Entropy, Generalised Linear Model, Generalised Additive Model) that gave the highest AUC values in a *k*-fold leave-one-out cross-validation test. Equivalent plots for all five bioclimate variables and all model algorithms are presented in S1 Fig in S1 File.

Our models for the swallow failed to achieve the high AUC scores found in our previous study of the bush-crow. This is unsurprising because of the lower quality of data available for the swallow. The range of the bush-crow can be very precisely delineated due to the bird's distinctive appearance and highly conspicuous nests, which are visible for up to a kilometre [12, 29, 53]. In contrast, the swallow is an unobtrusive, wide-ranging aerial forager, and its nests are usually concealed within huts [37]. For this reason, our training data contained very few 'known' absences for the swallow compared to the bush-crow, and this probably resulted in the differences in estimated model accuracy. Nonetheless, the similarity across different model algorithms in both the area of predicted occurrence and the importance of the dry-season precipitation and maximum temperature variables suggest that the overall result is robust.

Our projections of potential range reductions of the two species under a selection of GCMs, whilst varying in severity between models, are consistent across all predictions. The outcomes for both species are of conservation concern, with severe loss of potential range under all GCMs and RCPs, even as soon as 2050. Many scenarios, particularly for the bush-crow, indicate total loss of potential range by 2070. Even under RCP2.6, the most optimistic scenario which requires strong mitigation strategies to be employed urgently [52], the bush-crow is projected to lose 85% of its potential range by 2050 and 90% by 2070, and the swallow 56% and 68% in the same periods. For neither species did the models predict that any areas around their current ranges would become suitable, as they already occupy the coolest area in the region. Studies modelling changes in ranges and reporting rates of species (the latter being proxies for local abundance) often predict declines in both, indicating that models of range

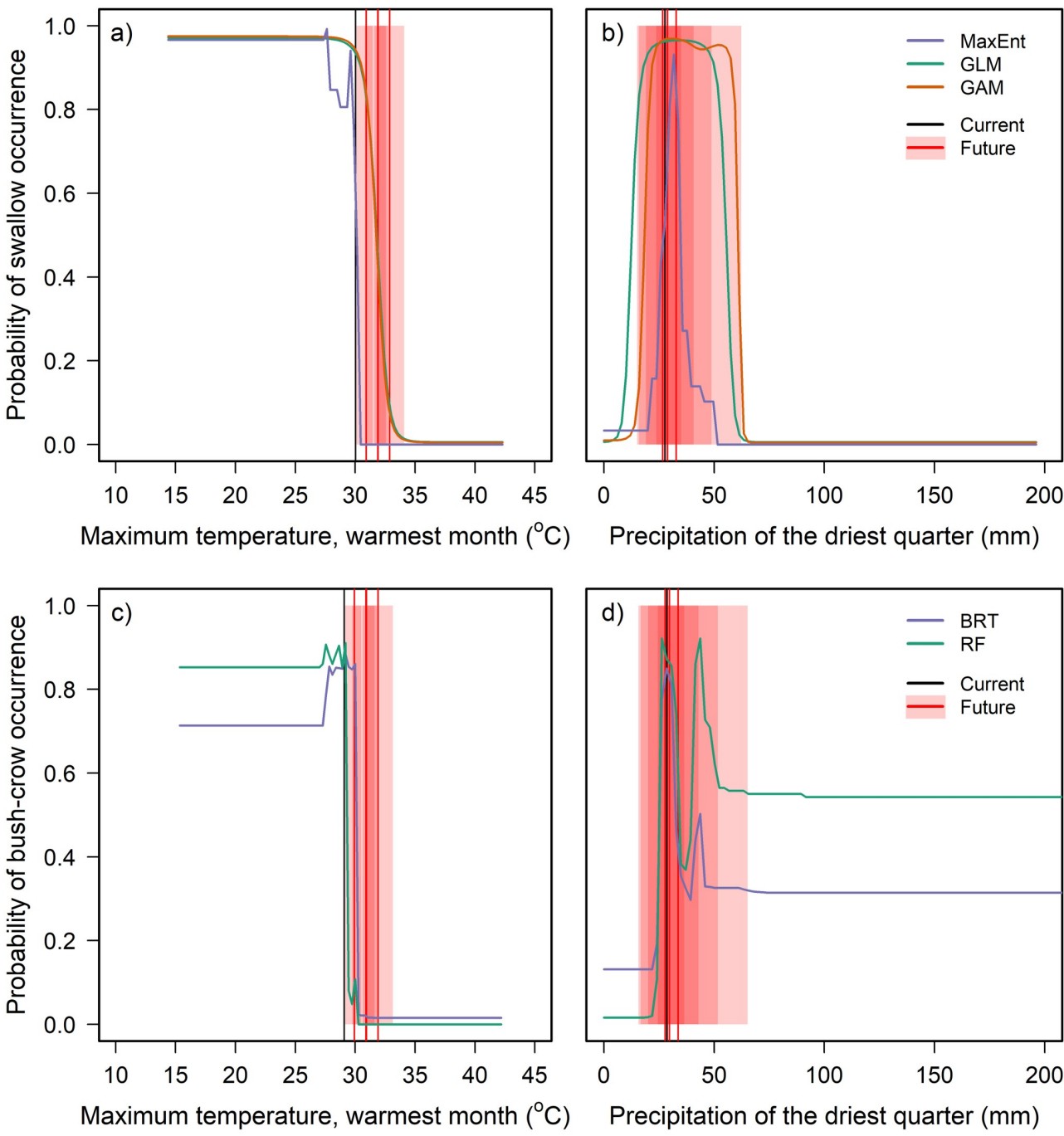

**Fig 4. The relationship of modelled partial probability of occurrence of a+b) White-tailed Swallow and c+d) Ethiopian Bush-crow to a+c) maximum temperature of the warmest month and b+d) precipitation of the driest quarter, compared to current and future projected mean values.** Black vertical lines indicate the current mean value of each bioclimate variable [41] within a convex hull fitted around each species' distribution. Red vertical lines show the projected mean value in 2070 for each of four Representative Concentration Pathways (RCP) (IPCC 2014). Red shading shows the range of projected mean values across six Global Circulation Models for each RCP, and appears darker where these ranges overlap (therefore corresponding to more likely scenarios). See Fig 3 for further details on response curves.

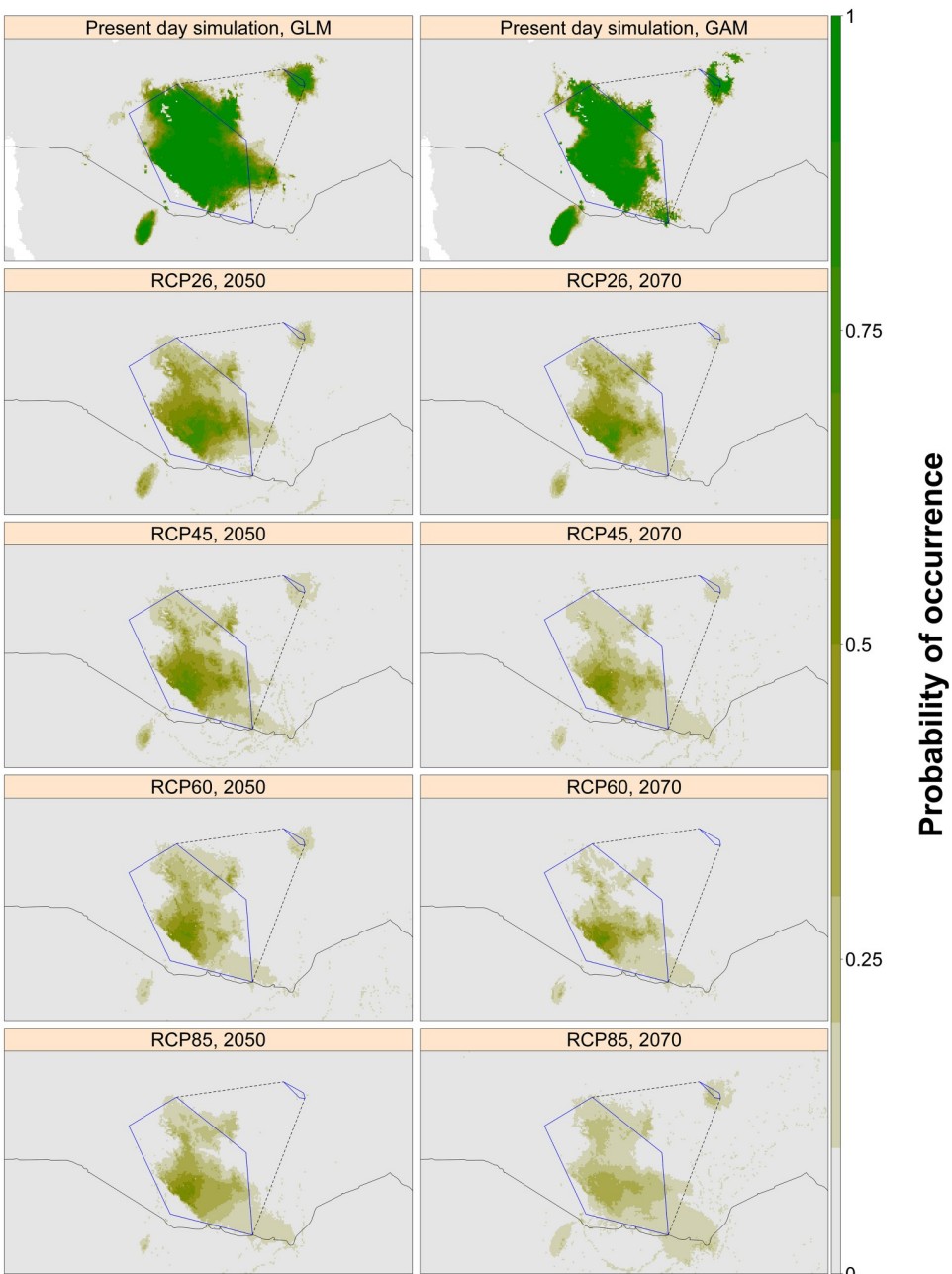

**Fig 5. Projected current and future potential range for the White-tailed Swallow, based on climate-only SDMs.**
Predictions for the four IPCC RCPs are presented in two future dates, 2050 and 2070. Each panel represents the mean probability of occurrence under each scenario, averaged across six GCMs and the three best-performing model algorithms (MaxEnt, GLM and GAM) under current conditions, according to *k*-fold LOOCV AUC. Dark green shows areas with a high probability of climatic suitability, fading through brown to grey, which shows areas with a low probability of climatic suitability. The blue polygon shows convex hulls fitted around the White-tailed Swallow's current distributions in the core range and on the Liben Plain, whilst the dashed line shows the complete hull if these two populations are considered to be continuous. International borders are plotted using the '*wrld_simpl*' dataset available in the '*maptools*' package in R [47, 48].

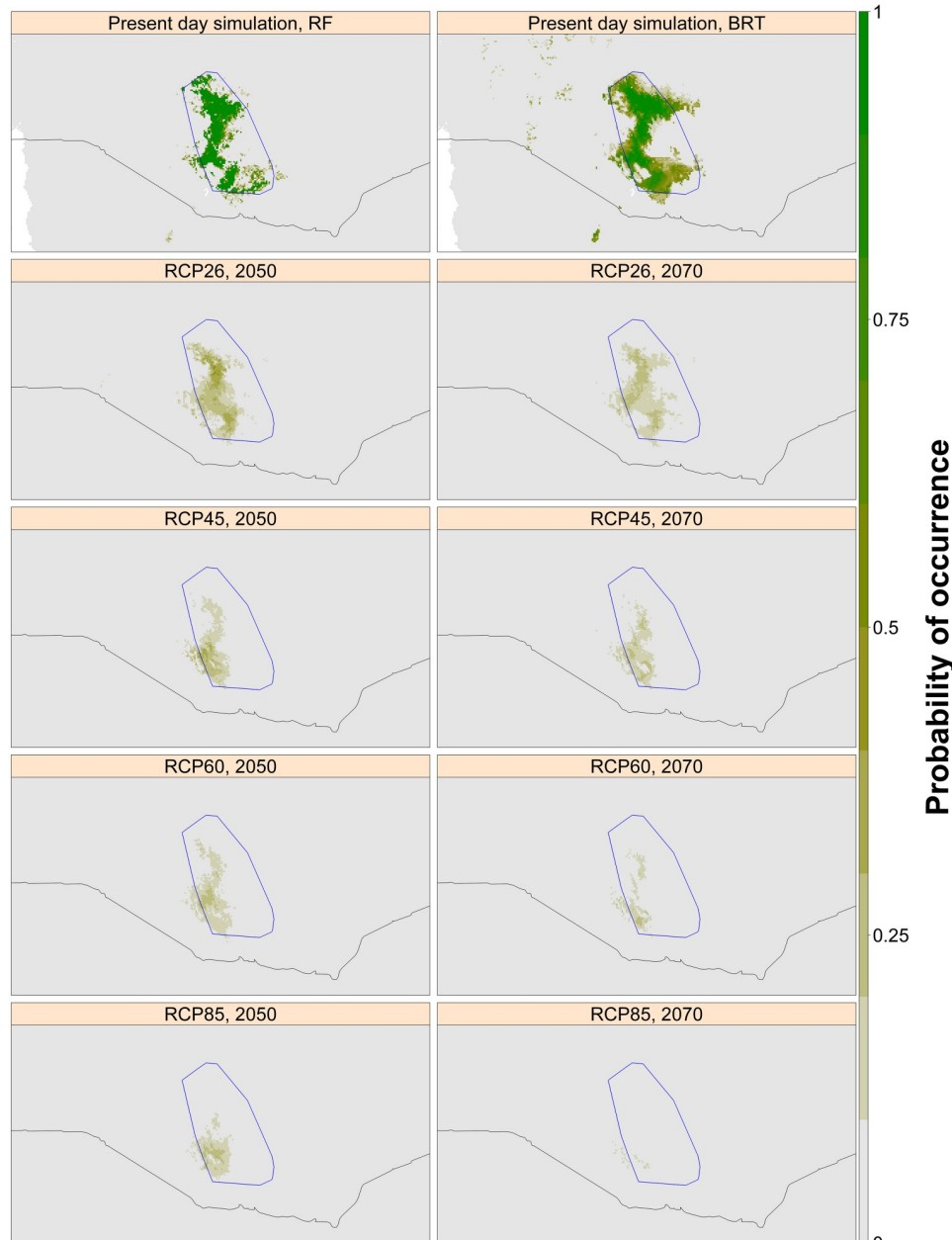

**Fig 6. Projected current and future potential range for the Ethiopian Bush-crow based on climate-only SDMs.**
Predictions for the four IPCC RCPs are presented for two future dates, 2050 and 2070. Each panel represents the mean probability of occurrence under each scenario, averaged across six GCMs and the two best-performing model algorithms (BRT and RF) under current conditions, according to *k*-fold LOOCV AUC. Dark green shows areas with a high probability of climatic suitability, fading through brown to grey, which shows areas with a low probability of climatic suitability. The blue polygon shows the convex hull fitted around the bush-crow's current distribution. International borders are plotted using the '*wrld_simpl*' dataset available in the '*maptools*' package in R [47, 48].

extent alone are likely to overestimate future population sizes [54]. It is therefore likely that decreases in population density within the remaining climatically suitable areas will also occur.

Despite the uncertainty inherent in species distribution modelling and the projection of potential ranges under future climate scenarios, numerous studies have found that climate

**Table 1. The area of mean current and future potential White-tailed Swallow and Ethiopian Bush-crow ranges based upon climate-only SDMs.**

| Scenario | White-tailed Swallow | | Ethiopian Bush-crow | |
|---|---|---|---|---|
| | Mean potential range / km$^2$ (min–max) | Mean percent left (min–max) | Mean potential range / km$^2$ (min–max) | Mean percent left (min–max) |
| Current | 8,311 (6,571–9,532) | - | 3,495 (3,384–3,606) | - |
| Year 2050, RCP 2.6 | 3,642 (0–8,568) | 44 (0–103) | 522 (0–1,988) | 15 (0–57) |
| Year 2050, RCP 4.5 | 2,218 (0–6,591) | 27 (0–79) | 263 (0–1,490) | 8 (0–43) |
| Year 2050, RCP 6.0 | 1,761 (0–6,753) | 21 (0–81) | 202 (0–1,959) | 6 (0–56) |
| Year 2050, RCP 8.5 | 1,713 (0–6,188) | 21 (0–74) | 136 (0–954) | 4 (0–27) |
| Year 2070, RCP 2.6 | 2,664 (0–6,380) | 32 (0–77) | 342 (0–2,160) | 10 (0–62) |
| Year 2070, RCP 4.5 | 1,583 (0–5,360) | 19 (0–64) | 173 (0–1,463) | 5 (0–42) |
| Year 2070, RCP 6.0 | 1,274 (0–4,631) | 15 (0–56) | 65 (0–373) | 2 (0–11) |
| Year 2070, RCP 8.5 | 1,367 (0–7,950) | 16 (0–96) | 5 (0–37) | 0 (0–1) |

Current ranges are the mean area of suitable climate according to the best two (bush-crow: [12]) and three (swallow) model algorithms. Future potential ranges are averaged across projections from the same algorithms and six GCMs under each Year/RCP scenario, with the range of values presented in parentheses. Percentages are calculated relative to the mean simulated current range.

envelope models fitted to species' present distributions can reliably predict future changes in range boundaries and population trends [4, 55–58]. The greatest uncertainty in projections tends to arise from the choice of modelling technique and baseline climate data used in model fitting, and of GCM used for model projections [23–25]. We investigated a suite of modelling techniques, and selected those with the highest predictive capacity under current conditions, preferring methods which performed well using $k$-fold leave-one-out cross-validation. As our results were validated by data independent of, and spatially separate from, those used for model fitting [12], we consider the models reasonably robust.

An additional source of uncertainty arises from the accuracy of the present-day distribution data. For the bush-crow, the high AUC scores achieved by the models when predicting the species' current distribution [12] reflects the quality of the underlying data, and suggests that future predictions are likely to be robust. Moreover, as part of a study not reported in this paper, we conducted six walked transects, 6–10 km long, at sites selected to be at the edge of the known geographical range of the bush-crow, as established by the data used in this paper. Each transect was placed so that it began within the known range and ended outside it. In all cases no bush-crows were detected in the portion of the transects lying outside the previously known range, adding further confidence to the accuracy of the present-day distribution data. For the swallow, the lower AUC scores create uncertainty in the present-day models, which is likely to reflect under-recording of the species' presence. When carried through to future projections, this could lead to under-predicting the area which will remain climatically suitable. However, even allowing for some uncertainty in the magnitude of the projected loss of suitable range, the direction of the response was consistent across models, and severe enough to warrant conservation concern.

Projections of species' future ranges made using climate-only models usually have higher apparent precision than those incorporating non-climate variables as well because, for all the

uncertainty in climate projections, the uncertainty in predictions of change in land cover and other human impacts is much larger [3]. While climatic factors accurately describe current bush-crow [12] and swallow occurrence, future projections of these models still represent the maximum potential distribution of each species under each climate scenario, with further restrictions imposed by habitat availability and human land-use [3, 29].

Documented extinctions implicating climate change have been driven by biotic interactions [59], and to date none appears to have been driven solely by temperature intolerance [7]. The Ethiopian Bush-crow may therefore be a rare example of an endothermic species directly limited by heat intolerance, making it particularly sensitive to direct effects of temperature change, compared with other species studied so far [12, 29, 60]. For the White-tailed Swallow, temperature may directly affect breeding success by inhibiting the adults' ability to provide food [40]; it certainly seems improbable that an aerial insectivore, with inter-continental migrants as congeners, should occupy such a restricted range because of limited food availability or breeding sites.

Given the strong responses of the bush-crow to direct impacts of temperature, management interventions to compensate for effects of climate change on its distribution and abundance would need to have a large effect. The same may be true for the swallow. Whether the two species can minimise the impact of rising temperatures within their current range through behavioural change seems improbable, although it is perhaps significant that bush-crow nests built on electricity pylons, which are taller than most available natural nest sites and hence may be cooler, were recently recorded for the first time [53]. "Assisted migration" [61] seems equally unfeasible. Our models failed to find any suitable climate within 150–400 km of the current range, indicating that translocations would need to move the species over large distances, into new environments and the ranges of species to which they have no prior exposure. For an aerial insectivore like the swallow, finding large areas with suitable habitat and temperatures projected to persist long into the future is a challenging prospect [62]. Possibly the omnivorous bush-crow [60] could be bred in captivity and released experimentally into candidate sites to increase the chances of success [62]. However, such actions would have to be carefully managed and monitored to avoid any negative impacts on native fauna from releasing a non-native, dietary generalist [60, 63].

Other species exhibiting strong responses to temperature have already suffered a reduction in range, indicating an inability to respond physiologically to rising temperatures [17]. Two African species, Rudd's Lark (*Heteromirafra ruddi*) and Botha's Lark (*Spizocorys fringillaris*), are already projected to suffer complete range loss by 2055 under two out of three GCMs considered [54]. The Ethiopian Bush-crow and White-tailed Swallow must be added to this list of species at high risk of extinction due to climate change within their native range. Both species could become model systems for furthering our understanding of species' distributions, testing our models' ability to predict future changes, and assessing whether there is scope for conservation interventions to reduce the negative impacts of climate change. These two species have particular benefits as model species: the range boundaries of the Ethiopian Bush-crow, and changes therein, can be very precisely identified due to its distinctive and highly visible nests, and the White-tailed Swallow appears to nest largely in inhabited huts, making them both relatively easy to find and well known to local people, and raising the possibility that changes in its range and population could be tracked using questionnaire surveys. Both species are already star attractions in a region home to five endemic birds [43, 64], and have the potential to become flagship species for the impacts of climate change on avian diversity in Africa.

## Supporting information

**S1 File.**
(DOCX)

## Acknowledgments

We are most grateful for White-tailed Swallow records from Simon Busutill, Yilma Dellelegn, Merid Gabremichael, Kai Gedeon, Steve Rooke and Claire Spottiswoode. We thank Kai Gedeon, Yilma Dellelegn, Samuel Jones, Solomon Desta, Birhanu Dessalegn, Gufu Kashina, Abduba Huka, Godana Safaro, Okotu Dida and Tsegaye Tesfaye for their support in preparing and conducting fieldwork. Mike Brooke and Steve Willis provided helpful discussions on the analyses.

## Author Contributions

**Conceptualization:** Andrew J. Bladon, Paul F. Donald, Nigel J. Collar, Rhys E. Green.

**Data curation:** Andrew J. Bladon, Jarso Denge, Galgalo Dadacha.

**Formal analysis:** Andrew J. Bladon.

**Funding acquisition:** Andrew J. Bladon, Paul F. Donald, Nigel J. Collar, Mengistu Wondafrash, Rhys E. Green.

**Investigation:** Andrew J. Bladon, Paul F. Donald, Jarso Denge, Galgalo Dadacha, Rhys E. Green.

**Methodology:** Andrew J. Bladon, Paul F. Donald, Rhys E. Green.

**Project administration:** Andrew J. Bladon, Paul F. Donald, Mengistu Wondafrash, Rhys E. Green.

**Resources:** Andrew J. Bladon, Mengistu Wondafrash.

**Software:** Andrew J. Bladon.

**Supervision:** Paul F. Donald, Nigel J. Collar, Mengistu Wondafrash, Rhys E. Green.

**Validation:** Andrew J. Bladon, Rhys E. Green.

**Visualization:** Andrew J. Bladon, Rhys E. Green.

**Writing – original draft:** Andrew J. Bladon.

**Writing – review & editing:** Andrew J. Bladon, Paul F. Donald, Nigel J. Collar, Jarso Denge, Galgalo Dadacha, Mengistu Wondafrash, Rhys E. Green.

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
