## [Decision Letter · Decision Letter 0]

5 Mar 2021

PONE-D-20-40516

Climatic change and extinction risk of two globally threatened Ethiopian endemic bird species

PLOS ONE

Dear Dr. Bladon,

Thank you for submitting your manuscript to PLOS ONE. After careful consideration, we feel that it has merit but does not fully meet PLOS ONE’s publication criteria as it currently stands. Therefore, we invite you to submit a revised version of the manuscript that addresses the points raised during the review process.

We look forward to receiving your revised manuscript.

Kind regards,

Antoni Margalida, Ph.D.

Academic Editor

PLOS ONE

Journal Requirements:

3. We note that Figures 1, 5, 6 in your submission contain map images which may be copyrighted. All PLOS content is published under the Creative Commons Attribution License (CC BY 4.0), which means that the manuscript, images, and Supporting Information files will be freely available online, and any third party is permitted to access, download, copy, distribute, and use these materials in any way, even commercially, with proper attribution. For these reasons, we cannot publish previously copyrighted maps or satellite images created using proprietary data, such as Google software (Google Maps, Street View, and Earth). For more information, see our copyright guidelines: http://journals.plos.org/plosone/s/licenses-and-copyright.

3.1.    You may seek permission from the original copyright holder of Figures 1, 5, 6 to publish the content specifically under the CC BY 4.0 license. 

3.2.    If you are unable to obtain permission from the original copyright holder to publish these figures under the CC BY 4.0 license or if the copyright holder’s requirements are incompatible with the CC BY 4.0 license, please either i) remove the figure or ii) supply a replacement figure that complies with the CC BY 4.0 license. Please check copyright information on all replacement figures and update the figure caption with source information. If applicable, please specify in the figure caption text when a figure is similar but not identical to the original image and is therefore for illustrative purposes only.

Reviewers' comments:

Reviewer's Responses to Questions

**Comments to the Author**

1. Is the manuscript technically sound, and do the data support the conclusions?

Reviewer #1: Yes

2. Has the statistical analysis been performed appropriately and rigorously? 

Reviewer #1: Yes

3. Have the authors made all data underlying the findings in their manuscript fully available?

Reviewer #1: Yes

4. Is the manuscript presented in an intelligible fashion and written in standard English?

Reviewer #1: Yes

5. Review Comments to the Author

Reviewer #1: The authors model the distribution of two highly endemic bird species, the White-tailed swallow and the Ethiopian Bush-crow, to predict the future distribution of these species in 2050 and 2070, assuming 4 different global-change scenarios. They find that the ranges of both species will considerably shrink, which will put them in high risk of extinction.

This is an interesting and timely study, presenting two examples of species, which will probably go extinct as a direct effect of the recent anthropogenic changes in climate. I find the study clear and convincing, yet I have a few questions which I address below.

Minor Comments

Lines 78-81: The number of the reference (Foden et al. 2013) is missing. It should be given at the end of this sentence.

Lines 92: The reference “Collar and Stuart (1985)” is missing in the reference list; so is the number of the reference.

Lines 134-135: Why are White-tailed swallows difficult to detect? I would assume that an aerial forager like a swallow can be easily detected while flying. Please explain.

Lines 136-137: Please, explain and justify this approach. Has this area be studied or monitored intensively? What is the probability that White-tailed swallows occur or do not occur in this area? Without further explanation, to me the delimitation of the area in which you sampled the pseudo-absence points seems arbitrary.

Lines 234-235: By which criterion did you consider that the models were indistinguishable? To me the result you obtained with these different approaches (Fig. 3, 4, S1) seem to vary considerably. Why didn’t you use only the result from the best model (=MaxEnt)? You should briefly discuss, to what extent the results from these models differed and why.

Lines 298-300: Could the similarity in climatic variables determining the ranges of these two species be partially due to limited sampling? If the presence/absence of both species were determined monitoring the same transects, climatic variation will be limited and may yield similar results for species living in the same area.

Lines 340-342: Another important source of error or uncertainty arises from the poor accuracy of present-date distribution, which may be caused by poor monitoring or monitoring in a too limited area. Could you please discuss to what extent, and in which way, this may have biased your results.

Lines 372-373: Would it be possible to provide more and cooler nesting sites to mitigate the effects of climate change on the White-tailed swallow?

6. PLOS authors have the option to publish the peer review history of their article (what does this mean?). If published, this will include your full peer review and any attached files.

Reviewer #1: No

---

## [Author Response · Author response to Decision Letter 0]

17 Mar 2021

We thank the reviewer for some very helpful comments on our manuscript. We have worked to address and include all of these, and detail our responses below.

Minor Comments

Lines 78-81: The number of the reference (Foden et al. 2013) is missing. It should be given at the end of this sentence.

Response: Thank you for spotting this, we have added the reference number.

Lines 92: The reference “Collar and Stuart (1985)” is missing in the reference list; so is the number of the reference.

Response: Thank you for spotting this, we have added the reference.

Lines 134-135: Why are White-tailed swallows difficult to detect? I would assume that an aerial forager like a swallow can be easily detected while flying. Please explain.

Response: This is a good point. White-tailed Swallows are small and occupy part-open habitat, either perching inconspicuously in vegetation or flying fast and low, so that they can be surprisingly easy to miss by an observer even on foot. We also note that they probably cover large areas when foraging and might congregate in areas with plentiful food. This would make their detection at any particular place in the geographical range less likely. Hence, we did not feel comfortable making the assumption that a lack of detection from a moving vehicle represented the true absence of the species from an area. We have added the following sentence to clarify this (line 137–143): “This is because a) there are a number of other swallow species found in the area (43), making positive identification from a moving vehicle unreliable, b) White-tailed Swallows are small and often occur singly or in pairs (37) and (c), if they are like other swallow species, they are likely to forage over a large area and may congregate in areas with plentiful food. These things all make it unreliable to assume that the failure to detect them at a particular place from a moving vehicle denotes a true absence of the species from the surrounding area”.

Lines 136-137: Please, explain and justify this approach. Has this area be studied or monitored intensively? What is the probability that White-tailed swallows occur or do not occur in this area? Without further explanation, to me the delimitation of the area in which you sampled the pseudo-absence points seems arbitrary.

Response: The extent of this panel was chosen to extend beyond the intensively studied region, and both species’ known range boundaries. This is particularly important for SDMs which are to be used for prediction, as it ensures that the models can account for what happens under environmental conditions well beyond what the species currently experiences. However, sampling too far away from the species’ ranges limits the applicability of the models to their current distribution. We have added the following sentence to explain this (line 146–150): “…and because it represents a pragmatic compromise between choosing an area large enough to ensure a range of environmental variables extending beyond the species’ known distribution – which is important for making predictions based on possible future scenarios – but small enough to make the models biologically relevant to a species with such a restricted range (44)”.

Lines 234-235: By which criterion did you consider that the models were indistinguishable? To me the result you obtained with these different approaches (Fig. 3, 4, S1) seem to vary considerably. Why didn’t you use only the result from the best model (=MaxEnt)? You should briefly discuss, to what extent the results from these models differed and why.

Response: Apologies, we realised that this was not clear. The three models were indistinguishable based on their k-fold leave-one-out cross-validation AUC scores. We have added this to the figure legend (line 254): “…which were indistinguishable based on their k-fold LOOCV AUC scores”. Because previous studies have shown that the choice of model algorithm can have a strong influence on model predictions, we felt that it was not robust to choose just one of three algorithms which had a similar performance, which is why we assessed variable importance and future climatic suitability for all three models. We have added a sentence to the Methods (lines 177–179) to explain this: “Since these three models had similar k-fold LOOCV AUC scores (see Results), we assessed variable importance and future climatic suitability based on all three models to avoid biasing our results towards a single model algorithm (24,25)”. We have also added a sentence to the Results (lines 245–247) on the key differences between the results from the three model algorithms: “The GLM and GAM models predicted White-tailed Swallow occurrence across a slightly wider range of dry-season rainfall values, and at slightly higher temperatures, than did the MaxEnt model (Figure 3)”.

Lines 298-300: Could the similarity in climatic variables determining the ranges of these two species be partially due to limited sampling? If the presence/absence of both species were determined monitoring the same transects, climatic variation will be limited and may yield similar results for species living in the same area.

Response: This is an excellent point, and one we had not properly considered. On reflection, we believe that our results are robust because the underlying data did differ quite substantially. Although we used the same walked transects, there were a large number of incidental records of both species which were not constrained to come from the same areas. Similarly, although the pseudo-absence points were taken from the same panel, they were selected separately for each species. Finally, the previously published Bush-crow models did include a lot of true absence points from the road transects, which we did not include for the White-tailed Swallow. We have not added any notes on this to the Discussion, as we felt it would break up the current text, but we are happy to add a sentence if the reviewer feels that this is important.

Lines 340-342: Another important source of error or uncertainty arises from the poor accuracy of present-date distribution, which may be caused by poor monitoring or monitoring in a too limited area. Could you please discuss to what extent, and in which way, this may have biased your results.

Response: Thank you, this is a good point. We have added a paragraph to the Discussion (lines 369–383) to cover this: “An additional source of uncertainty arises from the accuracy of the present-day distribution data. For the bush-crow, the high AUC scores achieved by the models when predicting the species’ current distribution (12) reflects the quality of the underlying data, and suggests that future predictions are likely to be robust. Moreover, as part of a study not reported in this paper, we conducted six walked transects, 6–10 km long, at sites selected to be at the edge of the known geographical range of the bush-crow, as established by the data used in this paper. Each transect was placed so that it began within the known range and ended outside it. In all cases no bush-crows were detected in the portion of the transects lying outside the previously known range, adding further confidence to the accuracy of the present-day distribution data. For the swallow, the lower AUC scores create uncertainty in the present-day models, which is likely to reflect under-recording of the species’ presence. When carried through to future projections, this could lead to under-predicting the area which will remain climatically suitable. However, even allowing for some uncertainty in the magnitude of the projected loss of suitable range, the direction of the response was consistent across models, and severe enough to warrant conservation concern.”.

Lines 372-373: Would it be possible to provide more and cooler nesting sites to mitigate the effects of climate change on the White-tailed swallow?

Response: Interestingly, our (unpublished) data suggest that nest temperatures are not the causal mechanism here. White-tailed Swallows do not nest in the coolest available locations, and nest survival correlates more strongly with ambient temperatures than with the temperatures experienced at the nest. We have added some text to the previous paragraph (lines 396–397) to hint at this: “For the White-tailed Swallow, temperature may directly affect breeding success by inhibiting the adults’ ability to provide food”. Although these results are available in a PhD thesis (referenced), we would prefer not to discuss them in too much detail here, as we may yet try to publish this work as well. However, we can add some more detail if the reviewer feels it would improve the Discussion.

---

## [Editor Report · Decision Letter 1]

23 Mar 2021

Climatic change and extinction risk of two globally threatened Ethiopian endemic bird species

PONE-D-20-40516R1

Dear Dr. Bladon,

We’re pleased to inform you that your manuscript has been judged scientifically suitable for publication and will be formally accepted for publication once it meets all outstanding technical requirements.

Kind regards,

Antoni Margalida, Ph.D.

Academic Editor

PLOS ONE
---

## [Editor Report · Acceptance letter]

28 Apr 2021

PONE-D-20-40516R1 

Climatic change and extinction risk of two globally threatened Ethiopian endemic bird species 

Dear Dr. Bladon:

I'm pleased to inform you that your manuscript has been deemed suitable for publication in PLOS ONE. Congratulations! Your manuscript is now with our production department. 

Kind regards, 

on behalf of

Dr. Antoni Margalida 

Academic Editor

PLOS ONE